# Upper Motor Neuron Disorders: Primary Lateral Sclerosis, Upper Motor Neuron Dominant Amyotrophic Lateral Sclerosis, and Hereditary Spastic Paraplegia

**DOI:** 10.3390/brainsci11050611

**Published:** 2021-05-11

**Authors:** Timothy Fullam, Jeffrey Statland

**Affiliations:** Department of Neurology, University of Kansas Medical Center, Kansas, KS 66160, USA; tfullam@kumc.edu

**Keywords:** primary lateral sclerosis, amyotrophic lateral sclerosis, hereditary spastic paraplegia

## Abstract

Following the exclusion of potentially reversible causes, the differential for those patients presenting with a predominant upper motor neuron syndrome includes primary lateral sclerosis (PLS), hereditary spastic paraplegia (HSP), or upper motor neuron dominant ALS (UMNdALS). Differentiation of these disorders in the early phases of disease remains challenging. While no single clinical or diagnostic tests is specific, there are several developing biomarkers and neuroimaging technologies which may help distinguish PLS from HSP and UMNdALS. Recent consensus diagnostic criteria and use of evolving technologies will allow more precise delineation of PLS from other upper motor neuron disorders and aid in the targeting of potentially disease-modifying therapeutics.

## 1. Introduction

Jean-Martin Charcot (1825–1893) and Wilhelm Erb (1840–1921) are credited with first describing a distinct clinical syndrome of upper motor neuron (UMN) tract degeneration in isolation with symptoms including spasticity, hyperreflexia, and mild weakness [1,2]. Many of the earliest described cases included cases of hereditary spastic paraplegia, amyotrophic lateral sclerosis, and underrecognized structural, infectious, or inflammatory etiologies for upper motor neuron dysfunction which have since become routinely diagnosed with the advent of advanced neuroimaging as well as genetic testing [2,3,4].

The core clinical features of a distinct clinical entity, primary lateral sclerosis (PLS), were first described in 1945 and included insidious onset, slow progression without plateau or remission, and examination findings limited to the pyramidal tracts without evidence of involvement of additional parts of the central nervous system [5]. While controversy remains as to whether there is a pathologic ‘gold standard’ to distinguish PLS from amyotrophic lateral sclerosis (ALS), especially those variants with a predominant upper motor neuron phenotype, the more benign clinical prognosis, as well as differing findings on neuroimaging and biomarker studies, continues to make this entity a clinical category of interest for further research and therapy development [2,4,6]. Several diagnostic criteria have been proposed with relative agreement on the core criteria of the presence of upper motor neuron dysfunction on exam, presentation most commonly in the legs, bulbar region, or mixed limb/bulbar regions, slow progression with ≥4 years from symptom onset, with age at diagnosis ≥ age 20, and lack of marked fasciculations, atrophy, sensory findings, or a family history. PLS remains a diagnosis of exclusion—and requires testing to exclude alternative diagnosis (e.g., EMG, MRI, etc.). The main point of differentiation between criteria is the time lapse between symptom onset and monitoring for development of lower motor neuron development [6]. While initial criteria proposed by Pringle recommended ≥3 years, later criteria by Singer and Gordon recommended ≥4 years and the COSMOS study in PLS required ≥5 years [3,7,8,9]. In general, the possibility of a false positive diagnosis or transition into ALS decreases with time. More recent consensus criteria have proposed a diagnosis of ‘probable PLS’ after 2 years of a pure upper motor neuron phenotype with ‘definite PLS’ diagnosed after 4 years [4]. The goal of more permissive and accepted clinical criteria is to allow for more uniform investigations into the histopathology and biomarkers related to PLS in order to more accurately distinguish this condition from other conditions presenting with upper motor neuron dysfunction as well as allow for targeted testing of therapeutics [2,4].

## 2. Clinical Presentation of Select Upper Motor Neuron Disorders

### 2.1. Primary Lateral Sclerosis

Primary lateral sclerosis is often considered on a spectrum of motor neuron disorders including those that are lower motor neuron (LMN) dominant (progressive muscular atrophy) on one end, ALS in the middle with various degrees of upper/lower motor neuron dysfunction, and the pure upper motor neuron phenotype of PLS making the opposite end of the spectrum. The pure upper motor neuron phenotype represents 1–4% of all patients with motor neuron disease [3,10,11,12].

The majority of cases present after age 20, with the only exception being a rare form of juvenile hereditary PLS associated with the alsin gene [7,13]. Average symptom onset is earlier than ALS, occurring in the 5th–6th decade as opposed to an average of 65 years for ALS, with a slight male predominance, similar to ALS [6,7]. Onset of symptoms is often insidious, with patients having a more difficult time discerning the exact onset of symptoms as compared to those with ALS. At initial onset, symptoms may only be noted with activities such as running or repetitive activities requiring more fluidity. At the time of initial evaluation, patients typically report stiffness of the extremities (especially the legs), clumsiness leading to falls, or bulbar symptoms, such as dysarthria, dysphagia, or emotional lability. Patients may report periods of gradual decline followed by apparent stabilization that may last several months [11,14]. With the more typical lower extremity onset (90% of cases), patients often seek medical attention once spasticity has progressed to a point where it is significantly impairing locomotion and perhaps increasing falls. While patients may describe weakness, this is more typically secondary to incoordination/imbalance due to spasticity. On examination they typically have grossly preserved strength, with any weakness typically being Medical Research Council (MRC) grade 4 or greater in an UMN pattern (i.e., extensor weakness in upper extremities, flexor weakness in lower extremities) [6,8,15]. The majority of patients who do not present with spastic paraparesis will present with corticobulbar dysfunction including speech changes (rate change prior to dysarthria), dysphagia (milder as compared to ALS with a lower rate of gastrostomy requirements), as well as pseudobulbar affect [14,15,16]. Other less common presentations include a hemiparetic (Mill’s variant) and upper extremity onset, although the majority of upper limb onset variants include some lower extremity or bulbar dysfunction with pure upper limb onset being more consistent with ALS [17,18,19,20]. While the paraparetic form of PLS has recently been shown to be associated with an earlier age of onset as compared the hemiparetic or bulbar onset forms, site of onset does not appear to predict a difference in survival [14,19]. Spread typically occurs from side to side, region to region, and ultimately results in spastic quadriparesis and bulbar dysfunction. One case series demonstrated an ascending progression from lower extremities to arms after 3.5 years and from lower extremities to bulbar onset after 5 years [6,8,14]. Average symptom duration varies from 7.2–14.5 years based on prior case series, longer than the typical average duration of 3–5 years quoted for ALS [7,8].

Clinical examination in PLS often demonstrates spasticity, hyperreflexia, and only mild weakness as previously noted which is symmetric in most cases with a lack of sensory findings. Early speech deficits tend to be related to rate of production rather than frank dysarthria. While pseudobulbar affect is well described in PLS, its presence at initial presentation is atypical and may predict future development of LMN signs [6,15,17]. Spasticity on initial examination is more common in PLS as compared to ALS (47% vs. 4%) and there should be no evidence of LMN involvement outside of possible thinning of the small muscles of the hands in older patients and those with a longer disease course (less than 2% of cases). Distal hand atrophy should not preferentially affect the lateral hand muscles such as in the ‘split hand syndrome’ described in ALS [18,21]. Occasionally, patients can present with apparent bradykinesia, poverty of movement, and gait instability, which can mimic atypical parkinsonism [8,22]. Atypical parkinsonism can be further suggested when disease onset is in a single limb and spreads ipsilaterally, a pattern often seen with parkinsonian disorders. The apparent bradykinesia of PLS can be distinguished from extrapyramidal causes due to a lack of fatigue or decrement on repeated finger tapping [22]. Additionally, none of the described cases of PLS presenting with parkinsonian features demonstrated levodopa responsiveness nor positive dopamine transporter (DaT) SPECT scanning [22,23]. While asymptomatic, saccadic breakdown of smooth pursuits and supranuclear paralysis has been described as a feature of PLS that is not seen in ALS [3,11]. Urinary symptoms (frequency) can be a complaint in one-third to one-half of patients later in the disease course [3,12,24]. While dementia is generally considered less common in PLS as compared to ALS, early studies had small sample sizes due to the rarity of PLS and most only utilized neuropsychological screening instruments. More recent dedicated studies on the neuropsychological profiles of patients with PLS have recognized similar changes as compared to ALS cohorts in the domains of executive, language, memory, and fluency [25,26,27]. In a cohort study of 30 patients with PLS who underwent formal neuropsychiatric testing, 57% had cognitive and behavioral changes as classified by the revised Strong consensus criteria with 17% meeting criteria for behavioral variant FTD [26,28].

### 2.2. Upper Motor Neuron Dominant ALS

Following publication of the Pringle Criteria for PLS in 1992, there was renewed interest in studies related to PLS. Unfortunately, while the criteria were widely used, there was a lack of specificity and many subsequent studies included patients in the umbrella term of PLS despite findings of LMN abnormalities clinically or by electromyography (EMG) [3,8,11,24]. New pathologic criteria for ALS, as well as case reports of late conversion from a PLS phenotype to ALS reinvigorated debate as to the utility of PLS as a distinct clinical entity as well as the appropriate time to wait until providing patients with a diagnosis of PLS [29,30,31,32,33,34]. These questions ultimately led to further case reviews of patients with an upper motor neuron dominant phenotype and proposition of the subcategory of UMN-dominant ALS. UMNdALS is defined as having symptoms lasting less than 4 years or disability secondary to UMN signs with known EMG denervation or LMN signs on exam that do not yet meet criteria for clinically definite, clinically probable, or probable-laboratory-supported ALS as defined by the revised El Escorial criteria [8,15,35].

UMNdALS patients show an intermediate time to death as compared to ALS, at 3–5 years, and PLS at 7.2–14.5 years [7,8,18]. EMG may be predictive of future development of clinical LMN signs within 6 months of identification and were identified within 4 years in 77% of patients who initially presented with a pure UMN phenotype [8]. Bulbar symptoms were more common in those with UMNdALS and ALS as compared to PLS patients across all visits and onset of disease in the bulbar region was also more common in UMNdALS (33%) and ALS (20%) as compared to PLS (11%) [15]. Additional findings of atrophy on initial examination, weight loss, and any MRC grade less than 4 at initial visit were predictive of UMNdALS or ALS as compared to patients remaining with a PLS phenotype. In a small comparison study of UMNdALS as compared to PLS and ALS, UMNdALS was distinguished clinically from ALS due to presentation with spasticity (100% UMNdALS vs. 70% ALS) as well as evidence of hyporeflexia or bulbar fasciculations on exam predictive of ALS [8,15,18].

### 2.3. Hereditary Spastic Paraparesis

Hereditary spastic paraplegia (HSP) is a syndrome characterized by bilateral lower extremity spasticity and weakness, which are a predominant manifestation for which genetic mutations are a causative factor. There are over 70 genetic subtypes identified which include all patterns of inheritance (autosomal dominant, autosomal recessive, X-linked, mitochondrial) [36,37]. Age of onset is variable given the various causative genetic and molecular mechanisms of disease; however, the more common autosomal dominant forms typically occur in adulthood between the second and third decades. Onset of symptoms prior to age 35 reliably distinguishes HSP from PLS. Unlike PLS, HSP is assumed to be genetic and diagnosis can span the lifespan from childhood onset to older age. Based on phenotype, HSP can be classified as ‘pure’ or ‘uncomplicated’ when the spastic paraplegia and subtle lower extremity dorsal column impairment is the primary manifestation or ‘complicated’ when additional neurologic or systemic abnormalities, such as dementia, seizures/epilepsy, ataxia, mental retardation, neuropathy, distal atrophy, or visual changes, co-exist. The most common form of autosomal dominant disease, SPG4 (spastin, 30–40% of autosomal dominant families) is typically associated with ‘uncomplicated’ forms of disease while others such as SPG11 (spatacsin, autosomal recessive, 50% of recessive families) are frequently associated with cognitive delays and characteristic neuroimaging findings (i.e., thin corpus callosum) [6,38,39]. A detailed review of all the various forms of HSP is beyond the scope of this article, however, [33] and [34] provide an in-depth overview.

While HSP, PLS, and ALS can at times be clinically indistinguishable, there are several factors which have been reported as helpful in identifying the underlying etiology. A positive family history would be more suggestive of HSP as ALS is only hereditary in 10% of cases. PLS is considered a sporadic disorder, with only a single autosomal dominant family reported and a rare juvenile form associated with the alsin gene, which is considered a separate entity [13,40]. HSP tends to affect both the descending corticospinal tracts as well as the ascending dorsal columns, therefore, diminished vibratory sensation on examination may help distinguish a clinical case of HSP from PLS or UMNdALS [41]. In addition to loss of vibratory sensation, given the long-standing nature of the disorder, patients are more likely to have notable foot deformities as compared to those with PLS/ALS. While reflexes may be brisk in the upper extremities in uncomplicated HSP, there tends to be a lack of significant weakness or loss of dexterity in the upper extremities in uncomplicated HSP which can serve as a useful distinguishing feature. Symmetry at presentation is supportive of both HSP and PLS and would be uncommon for ALS [3,8,41]. Despite the above clinical features, genetic testing remains essential for ruling out HSP as the etiology for an apparently sporadic adult onset UMN syndrome with leg onset. A prior large cohort study of sporadic adult-onset UMN syndromes, including those with SPG4 or SPG7 mutations, demonstrated significant overlap between cohorts in age of onset, urinary urgency, and evidence of dorsal column impairment. The only potential supportive/predictive findings for distinguishing PLS from HSP in this study were onset of symptoms in the arms or bulbar region or development of marked asymmetry throughout the disease course being more predictive of PLS as compared to HSP. Ultimately, it is the evolution over time that best distinguishes the disorders clinically as PLS will progress to other regions and is more likely to demonstrate asymmetry [42].

Key clinical characteristics between select upper motor neuron disorders are summarized in Table 1.

## 3. Diagnostics

Upon recognition of a predominant upper motor neuron syndrome through a thorough history and examination, the most challenging aspect of diagnosis is distinguishing between PLS and UMNdALS or potentially HSP. Structural, infectious, and demyelinating diseases that may mimic the above upper motor neuron syndromes can often be effectively ruled out through neuroimaging (MRI) of the brain and spine in addition to basic laboratory studies. Basic chemistries, serum B12 and copper, HIV testing, and potentially HTLV 1/2 and paraneoplastic workup in certain clinical scenarios are often considered. Routine CSF testing may also be appropriate in atypical presentations, such as those that present sub-acutely or with altered mentation. CK is not typically elevated in those patients with PLS or HSP as compared to ALS, however, this is not a reliable clinical marker to distinguish these conditions [2,4,6,7]. The following sections outline some of the key differences between the upper motor neuron syndromes in routine clinical testing as well as emerging technology to distinguish the conditions.

### 3.1. Electromyography

Given the potential for late development of lower motor neuron involvement, which is often able to be identified by electromyography (EMG) prior to clinical manifestation, EMG remains a core method to distinguish PLS from UMNdALS. The initial study by Gordon et al. that proposed the diagnostic category of UMNdALS defined clinically pure PLS as those who had a normal EMG of three limbs, bulbar, and paraspinal muscles while those with UMNdALS had evidence of lower motor neuron weakness, wasting, or fasciculations limited to one to two muscles or minor denervation on EMG, including sparse fibrillation/positive sharp waves, or minor motor unit potential remodeling in one to two muscles not meeting revised El Escorial criteria for any ALS category [8]. In the initial studies, EMG abnormalities occurred after a median of 3.17 years after development of UMN signs and preceded the LMN signs (atrophy, fasciculations, weakness) by six months. The clinical utility of identifying UMNdALS was to identify patients with UMN predominant disability that also had LMN signs as this predicted a prognosis, both in terms of survival and disability, that was intermediate as compared to PLS and typical ALS [8,15]. While subsequent studies have reported no difference in survival between those with a diagnosis of PLS and a normal EMG and those with minimal denervation changes restricted to a single muscle, these studies did not include the presence of LMN signs in their definitions and limited the findings to a either ‘minimal’ EMG changes to a single muscle or fasciculations limited to no more than two muscles or fibrillation or motor unit changes in any muscle as qualifying as minor denervation on EMG [43,44]. A prior study by Singer et al. did note increased lower limb disability in those PLS patients with active denervation changes on their EMGs as compared to those patients with a normal EMG, although, the study was limited by a small sample of 25 total patients [12]. Taken together, if denervation changes are limited to a single or few muscles and are non-progressive or disappear on a repeat EMG at 4 years in an otherwise pure UMN syndrome, a diagnosis of definite PLS can be made. On the other hand, if a patient has evidence of denervation on their initial EMG but they develop limited/focal LMN symptoms (weakness, fasciculations, atrophy) over the course of 4 years but still do not meet revised El Escorial or Awaji criteria for ALS, a diagnosis of UMNdALS as opposed to probable PLS would be appropriate as they may have an intermediate prognosis as compared to typical PLS [4].

Various cut-offs for the development of lower motor neuron findings on EMG to distinguish PLS from UMNdALS or ALS have been used, with the Pringle criteria using 3 years, the Gordon and Singer criteria 4 years, and the COSMOS study using 5 years. One common factor throughout most criteria is the allowance of rare/occasional increased insertional activity in one to a few muscles which should be non-progressive over time [3,4,7,8,9]. After four years, the probability of those with absent or sparse LMN findings developing new LMN findings is low (approximately 20%) and there appears to be no difference in clinical symptoms, disability, or outcome measures between those with normal EMGs or minor denervation at initial visit [8,44]. Given the above and the desire to enroll patients into future clinical trials at an earlier stage of disease, recent consensus criteria have proposed a ‘probable PLS’ category for those with a lack of LMN findings for 2–4 years from diagnosis and a ‘definite PLS’ category for those with lack of active LMN degeneration 4 or more years from diagnosis [4].

More advanced techniques such as motor unit number estimation (MUNE) have been shown to be significantly reduced in those with ALS as compared to PLS, who typically show normal or mild reduction in motor unit hand muscles [6,45,46,47]. Additionally, single fiber EMG has demonstrated increased jitter in both patients with HSP and PLS, however, jitter was more prominent in and blocking restricted to HSP patients [48]. Finally, while abnormal SSEPs are typically thought to be more commonly seen in those with HSP as compared to PLS, abnormal leg SSEPs were not able to reliably distinguish 14 patients with genetically confirmed HSP from 90 patients with a sporadic UMN syndrome, 78 of which had onset of symptoms in their legs [42].

### 3.2. Genetic Testing

PLS is considered an adult-onset sporadic disorder, separate from the juvenile-onset PLS-like syndrome associated with the alsin gene which is inherited in a recessive manner [4,13]. Outside of a recently reported Canadian kindred with dominantly inherited PLS like syndrome with progressive nonfluent aphasia, no consistent genetic basis for PLS has emerged [49]. Testing for C9orf72, the most common hereditary cause of ALS in Caucasian populations, is reasonable to consider in the workup for HSP and has identified mutations in 0.9–2.9% of PLS patients as compared to identification in 30–40% of familial ALS patients and 5–10% of sporadic ALS cases [43,50,51]. Screening panels for HSP is reasonable in those with a progressive UMN syndrome initially restricted to the lower extremities, even if there is no family history, as 20–40% of HSP cases may lack a clear family history [2,41]. Over 70 genes have been described as associated with HSP with various forms of inheritance (autosomal dominant, autosomal recessive, X-linked). As commercial testing has become more affordable, those with lower extremity onset UMN syndromes should be considered for both C9orf72 and HSP testing. Given the current availability of gene targeted clinical trials for C9orf72 mutations, screening for this mutation in all pure UMN syndromes may be reasonable as well [6,39,41].

### 3.3. MRI

In addition to ruling out structural/inflammatory/infectious etiologies for UMN syndromes, quantitative MRI analysis has identified additional ways to potentially distinguish PLS from other UMN syndromes. Focal ‘knife edge’ atrophy of the precentral gyrus has typically been identified as the only structural abnormality allowed in PLS. The focal atrophy of the precentral gyrus is typically greater than that found in ALS and is felt to be secondary to the duration of the disease [3,52]. While T2 hyperintensity within the corticospinal tracts has been described in both ALS and PLS, the timing of the finding may be relevant [3,6]. A recent study identified those with UMNdALS with positive corticospinal tract hyperintensity (CST+) on T2-weighted conventional MRI and compared these patients to those with UMNdALS without corticospinal tract hyperintensity, classic ALS patients, and those with ALS-FTD disorder. Those patients with CST+ UMNdALS had a shorter duration of symptoms prior to MRI and a disease progression rate that was three times higher as compared to those with UMNdALS without CST+. Given that the MRI in the UMNdALS CST+ was typically obtained 9.6 ± 5.5 months from symptom onset, the study suggests that CST+ within the first 1–2 years of symptoms in a patient with a pure UMN phenotype may portend a worse prognosis and help distinguish UMNdALS from PLS [53]. Despite its potential utility in predicting prognosis in upper motor neuron disorders, T2 CST+ may not be as useful from a diagnostic perspective as it has also been demonstrated in a normal population without evidence of motor neuron disease, specifically in those over the age of 50 [54].

Finegan and colleagues recently completed a study using volumetric and diffusion tensor imaging of patients with definite PLS as compared to probable PLS based on the 2020 consensus criteria. The study noted motor cortex atrophy in early PLS as compared to healthy controls and was able to identify less severe and more focal structural alterations in probable PLS as compared to definite PLS. Additionally, there were significant and symmetric white matter abnormalities in the corticospinal tracts and corpus callosum on DTI in the definite PLS group that were not seen in the probable PLS group suggesting that the gray matter pathology in PLS precedes the changes in white matter degeneration. This differs from ALS where corticospinal tract and corpus callosum degeneration has been shown to be an earlier feature on neuroimaging with gray matter abnormalities appearing later [55]. Prior diffusion tensor imaging studies comparing PLS to ALS have supported the hypothesis of mid-body corpus callosum white matter changes in PLS reflecting secondary Wallerian degeneration as the result of neuronal death in the primary and premotor cortices. While there is similar distal CST involvement in PLS and ALS patients, those with PLS demonstrated more severe damage to the rostral portions of the CST as well as the motor callosal fibers as compared to ALS patients [56]. These findings support the recent PLS consensus classification system and argues for earlier intervention/inclusion into clinical trials [55,57,58,59]. Another volumetric study comparing 100 patients with ALS with 33 PLS patients and 100 controls found significant volumetric differences between subgroups with PLS, demonstrating unique volumetric losses in the thalamic sensory nuclei and lateral geniculate region and sparing of the amygdala as compared to ALS patients [60]. In addition to hyperintensity of the CST, HSP may demonstrate several unique findings identified on conventional MRI such as thinning of the corpus callosum in SPG11 or diverse findings such as leukodystrophy, hypomyelination, spinal cord or cerebellar atrophy, or hydrocephalus in complex forms of HSP [39,41].

### 3.4. Positron Emission Tomography

Flumazenil positron-emission tomography (PET) binds to GABAa receptors with a decrease in binding indicating greater neuronal loss/dysfunction in the affected region. Prior studies have identified decreased binding in PLS within the motor cortex bilaterally as compared to controls, similar to prior studies comparing sporadic ALS to controls. While there were some areas of relative further decrease in flumazenil PET binding in the anterior frontal and orbito-frontal regions in ALS patients as compared to PLS, the sensitivity/specificity of this finding to distinguish PLS from ALS has yet to be confirmed [2,6,61]. Similarly, the fluorodeoxyglucose (FDG) PET ‘stripe sign’ of focal motor cortex hypometabolism in PLS as compared to diffuse frontal cortical hypometabolism in ALS patients has not shown significant sensitivity/specificity for distinguishing UMNdALS from PLS, specifically [3,62].

### 3.5. Transcranial Magnetic Stimulation

Threshold tracking transcranial magnetic stimulation techniques have demonstrated cortical dysfunction across motor neuron diseases which exists on a spectrum of cortical inexcitability as a predominant feature in PLS and cortical hyperexcitability as a predominant feature in ALS [27,63,64]. The stimulus intensity required to evoke a response has been shown to be higher in PLS as compared to ALS patients on average whereas cortical excitability is preserved in cases of HSP [64]. In addition to distinguishing between cases of PLS and ALS based on differences in cortical excitability, TMS threshold tracking techniques may also help differentiate PLS from HSP, which has persevered cortical excitability as compared to controls [64]. PLS patients tend to show unobtainable or highly delayed central motor conduction in the affected limbs as compared to ALS patients [4,17,65].

### 3.6. Neurofilaments and Chitinases

Candidate biomarkers are an area of active research to exploit CSF or serum elements which may provide better diagnostic and/or prognostic assessments of neurodegenerative disease, including motor neuron spectrum disorders. Both neurofilament light chains (NfL) and phosphorylated neurofilament heavy chains (pNfH) can be measured in serum and in CSF and have been shown to be correlated with disease severity and survival parameters in motor neuron disease [66,67]. The mechanism underlying their elevation (UMN degeneration versus LMN degeneration) continues to be a matter of debate. One recent study speculated that elevated NfL in motor neuron disease was reflective of both the overall rate of motor neuron degeneration as well as the amount of degeneration in the anterior horn cells. This is supported by lower comparative values in pure UMN degeneration syndromes such as PLS or HSP as compared to classic ALS, although, diagnostic cut-offs have yet to be clearly defined [68,69,70,71]. Alternatively, recent studies have found higher levels of CSF NfL and serum pNfH chains in typical ALS as well as those with UMNdALS and pseudobulbar palsy as compared to those syndromes that primarily affect the lower motor neurons such as progressive muscular atrophy or the flail arm/leg syndromes [66,67]. These findings along with the known elevated levels (serum or CSF) of NfL or pNfH in ALS as compared to PLS or HSP may suggest that the rate of degeneration of the corticospinal tract drives the differences between these disorders [66,67,72,73]. CSF levels of chitinase proteins, a marker for microglial activation, have shown some promise in that they are elevated in patients with ALS as compared to PLS, however, the fact that they do not outperform NfL suggests their utility may be in monitoring therapies targeting microglial activation [74,75].

### 3.7. Histopathological Findings

The earliest published autopsy studies in the modern era of patients with PLS (1977–1997) occurred prior to the discovery of Bunina bodies or ubiquitinated neuronal inclusions including transactive response DNA binding protein-43 (TDP-43) in 1997. These early studies highlighted atrophy of the precentral gyrus, degeneration of the corticospinal tract through the spinal cord, and loss of Betz cells in the precentral gyrus [3,7,76,77,78]. Those studies after 1997 continue to highlight loss of Betz cells in the primary motor cortex and corticospinal tract degeneration, however, many were complicated cases that included dementia or parkinsonian features with mention of either Bunina bodies or ubiquitinated positive inclusions which makes it hard to conclude that these were true PLS cases and not on the spectrum of ALS overlap [7,79]. Several cases were primarily published based on novel clinical-pathologic associations due to their combination of upper motor neuron degeneration with atypical parkinsonian syndromes or common dementias such as FTD or dementia with Lewy bodies, making them difficult to generalize to more ‘clinically pure’ cases of PLS [80,81,82,83,84,85]. A recent autopsy study of seven cases that met 2020 consensus criteria for PLS demonstrated TDP-43 neuronal cytoplasmic inclusions (NCI) in the primary motor cortex of all five cases that it was available for analysis. Additionally, all cases demonstrated chronic degeneration of the corticospinal tract and no obvious reduction in the number of lower motor neurons in the hypoglossal nucleus or ventral gray matter but rare TDP-43 NCI in the LMNs. The number of LMN NCI in the PLS cases was estimated to be less than 1% as compared to an average of approximately 20% of cells demonstrating NCIs in ALS cases, including those designated as ‘early ALS’ [86]. While the frequency of TDP-43 NCI in the UMN of PLS may explain the association of FTD with both disorders, the significant difference in LMN TDP-43 NCI in the PLS cases as compared to ALS (both early and late) suggests that the disorders differ in their degree of LMN vulnerability [86]. TDP-43 pathology in ALS has been shown to be more predominant in motor rather than extra-motor neurons, with additional presence noted in glial cells. Lesions tend to be skein-like while the rare cases of TDP-43 identified in PLS patients were described as rod shaped [87]. Conclusions on definitive histopathological findings to distinguish PLS from ALS remains elusive due to limited case reports in the modern era of immunohistochemistry [4]. Findings in HSP include degeneration of the distal axons in the descending corticospinal tract, maximal in the thoracic spinal cord. In addition, one can see degeneration of the ascending dorsal column fibers in the fasciculus gracilis, which is maximal at the cervicomedullary region. Both findings support a primary axonal dying back phenomenon. Betz cells have been described as reduced in HSP, similar to PLS [88].

Key diagnostic differences between ALS, HSP, and PLS are summarized below in Table 2.

## 4. Future Directions

Unfortunately, given its reliance on being clinically defined and its long trajectory, PLS has continued to be neglected in terms of clinical trial and therapy development. Recent consensus criteria with ‘probable’ and ‘definite’ PLS categories along with emerging neuroimaging, electrophysiologic, genetic, and molecular studies to distinguish PLS from UMNdALS and HSP will help clear the path for reliable clinical trial development, both for disease modifying as well as symptomatic therapy. Until such technology becomes more reliable, distinguishing PLS and ALS from UMNdALS remains important as this cohort of patients have been shown to have an intermediate progression in terms of disability and survival as compared to either PLS and ALS and may confound results of future clinical trials if they are included in PLS or ALS patient cohorts. As the costs of genetic testing continue to lower and more clinical trials are targeted towards specific gene mutations in both ALS and HSP, genetic testing will likely become a more routine part of the workup for pure upper motor neuron disorders. Despite being considered on the spectrum of ALS, PLS clearly has a longer survival and much slower rate of decline, as much as 30 times slower than that of ALS when measured by the ALS functional rating scale-revised (ALSFRS-R). More sensitive clinical trial tools, such as the development of a unique PLS functional rating scale and advanced MRI imaging techniques, may help encourage therapeutic development for PLS therapies and help shorten the time required to study these interventions [9,43,89].

## Figures and Tables

**Table 1 brainsci-11-00611-t001:** Key clinical differences between PLS, UMNdALS, ALS, and HSP.

Feature	PLS	UMNdALS	ALS	HSP
Incidence (per 100,000/year)	<0.1	1–3	1–3	4.3–9.6 ^1^
Age of onset (mean)	50	50	65	30–40
Gender Predominance	2–4:1 ^2^	1.6:1	1.6:1	1:1
Prognosis	Slowly progressive, decade or greater survival	Typical survival beyond a decade	3–5 years	Very slowly progressive, variable
Site of first symptoms	Lower limb 90%/Bulbar 10%	Lower Limb 30%/Bulbar 30%/Upper limb 30%	Lower Limb 30%/Bulbar 30%/Upper limb 30%	Lower Limb
Lowest MRC on initial evaluation ≤4?	-	+/-	+	-
Symmetry	Symmetric ^3^	Asymmetric	Asymmetric	Symmetric
Weight Loss at Diagnosis?	-	+	++	-
Urinary Frequency	++ (late)	+/-	-	++
Cognitive Impairment	+	+	++	+/-
Family History	None	10%	10%	40–60%
Spasticity	+++	+++	+/-	+++
PEG/Ventilator needs?	+/-	+	++	-
Frontotemporal Dementia	+	+	++	+/-
Pseudobulbar Affect	++	+	+	-

^1^ prevalence per 100,000. ^2^ Drawn from small populations, some suggest range may be closer to that observed in ALS. ^3^ Asymmetric features may help distinguish PLS from HSP, however.

**Table 2 brainsci-11-00611-t002:** Key diagnostic differences between ALS, HSP, and PLS.

Diagnostic Test	PLS	ALS	HSP
Denervation Potentials on EMG	+/- ^1^	+++	+/-
Abnormal SSEPs	+/-	-	++
Precentral Gyrus Atrophy (MRI)	++	+	-
Corticospinal Tract Hyperintensity on T2-weighted imaging	+ (late)	+ (early) ^2^	+
FDG-PET Findings	Focal motor cortex hypometabolism (‘stripe sign’)	Motor cortex + diffuse frontal cortex hypometabolism	Heterogenous areas of cortical hypometabolism
TMS MEPs prolonged? (early/late)	+++/+++	+/++	-/-
Neurofilament light chain levels	++	+++	+
Bunina Bodies	+/-	++	-
TDP-43 Inclusions	+ ^3^	++	-

^1^ one to few muscles, non-progressive over time. ^2^ Identification in first 1–2 years of symptom onset may help distinguish UMNdALS from PLS. ^3^ TDP-43 NCI reported in primary motor cortex of PLS patients but rarely found in LMNs as compared to ALS.

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
