# Peer review of "Upper Motor Neuron Disorders: Primary Lateral Sclerosis, Upper Motor Neuron Dominant Amyotrophic Lateral Sclerosis, and Hereditary Spastic Paraplegia"

_brainsci, 2021, doi:10.3390/brainsci11050611_

Round 1
Reviewer 1 Report
The review is well written and comprehensive. It doesn’t add too much new information on the topic above and beyond the recent PLS consensus criteria paper (Turner M, Talbot K Prac Neurol 2020). However, the sections on EMG, genetics and MRI for diagnosis of PLS were informative. In particular, the recent finding that hyperintensity of corticospinal tracts on MRI in those with UMNdALS could signal faster disease progression/worse prognosis is interesting. I thought the tables were well done.
The cognitive involvement of the groups could be expanded.
The use of UMNdALS versus PLS and ALS remains difficult and perhaps the authors can critically analyse these groups ie whether UMNdALS is needed and why. Perhaps in the future directions this could be expanded?
minor
31- of
48 core
206 over time
239 started
287 PET
Author Response
Timothy Fullam, M.D.
Maj, USAF, MC
University of Kansas Medical Center
3901 Rainbow Blvd
Kansas City, KS, USA 66160
717-576-5552
tfullam@kumc.edue
Drs. P. Hande Ozdinler and Bradley Turner
Guest Editors
Journal of Brain Sciences: Upper Motor Neurons in Health and Disease
May 2, 2021
Dear Drs. Ozdinler and Turner,
I am pleased to re-submit our review article entitled “Upper Motor Neuron Disorders: Primary Lateral Sclerosis, Upper Motor Neuron Dominant Amyotrophic Lateral Sclerosis, and Hereditary Spastic Paraplegia” by Dr. Jeffrey Statland and myself for consideration for publication in the Journal of Brain Sciences special issue Upper Motor Neurons in Health and Disease. We have incorporated the recommended changes as outlined below:
Major: (original comments/suggestions in italics)
P2.1.
‘Authors should provide a reference. Moreover, the less common upper limb onset should be commented, being widely more common in ALS than in PLS.’ AND
‘Even if different onset sites do not differ for survival, a recent study compared a younger age at onset for paraparetic PLS sub-phenotype compared to bulbar and Mills phenotypes (Hassan et al 2021).’
-Lines 80-85: Included references and modified text to read ‘Other less common presentations include a hemiparetic (Mill’s variant) and upper extremity onset, although the majority of upper limb onset variants include some lower extremity or bulbar dysfunction with pure upper limb onset being more consistent with ALS [17-20]. While the paraparetic form of PLS has recently been shown to be associated with an earlier age of onset as compared the hemiparetic or bulbar onset forms, site of onset does not appear to predict a difference in survival [14, 19].’
-Lines 85-86: Minor grammatical corrections to help with the addition of the above lines (capitalize ‘Spread’ to start new sentence, change ‘occurring’ to ‘occurs’, change ‘resulting’ to ‘results’)
-Lines 89-90- removed ‘ Other patterns of progression include bulbar to lower extremities followed by upper extremity involvement or hemiparesis/Mill’s variant’ (redundant given addition above)
-the above modification includes the recommended reference
‘Extra motor features such as parkinsonism should also be discussed (e.g. Parkinsonism and PLS: norlinah iM, Bhatia KP, Østergaard K, et al. Primary lateral sclerosis mimicking atypical parkinsonism. Mov Disord 2007;22:2057–62; Mabuchi n, Watanabe h, atsuta n, et al. Primary lateral sclerosis presenting parkinsonian symptoms without nigrostriatal involvement. J Neurol Neurosurg Psychiatry 2004;75:1768–71)’
-Lines 102-109: added ‘Occasionally, patients can present with apparent bradykinesia, poverty of movement, and gait instability which can mimic atypical parkinsonism [8, 22]. Atypical parkinsonism can be further suggested when disease onset is in a single limb and spreads ipsilaterally, a pattern often seen with parkinsonian disorders. The apparent bradykinesia of PLS can be distinguished from extrapyramidal causes due to a lack of fatigue or decrement on repeated finger tapping [22]. Additionally, none of the described cases of PLS presenting with parkinsonian features demonstrated levodopa responsiveness nor positive dopamine transporter (DaT) SPECT scanning [22, 23].’
-the above addition includes the recommended references
‘The cognitive involvement of the groups could be expanded’
-Lines 112-119: Expanded section discussing cognitive changes in PLS to include ‘While dementia is generally considered less common in PLS as compared to ALS, early studies had small sample sizes due to the rarity of PLS and most only utilized neuropsychological screening instruments. More recent dedicated studies on the neuropsychological profiles of patients with PLS have recognized similar changes as compared to ALS cohorts in the domains of executive, language, memory, and fluency [25-27]. In a cohort study of 30 patients with PLS who underwent formal neuropsychiatric testing, 57% had cognitive and behavioral changes as classified by the revised Strong consensus criteria with 17% meeting criteria for behavioral variant FTD [26, 28].’
-Lines 381-384: Included discussion of TDP-43 pathology in and ALS and PLS and the potential link to FTD ‘While the frequency of TDP-43 NCI in the UMN of PLS may explain the association of FTD with both disorders, the significant difference in LMN TDP-43 NCI in the PLS cases as compared to ALS (both early and late) suggests that the disorders differ in their degree of LMN vulnerability [86].’
P2.2.
It would be useful to clear the point about “EMG denervation or LMN signs that do not yet meet criteria for ALS”, referring to revised El Escorial and/or Awaji criteria.
-Lines 129-133: clarified by changing final sentence of paragraph to ‘UMNdALS is defined as having symptoms lasting less than 4 years or disability secondary to UMN signs with known EMG denervation or LMN signs on exam that do not yet meet criteria for clinically definite, clinically probable, or probable-laboratory-supported ALS as defined by the revised El Escorial criteria [8, 15, 35].’
Lines 121-126. Please provide a reference.
-prior lines 121-126 are now 135-140 and references have been included ‘EMG may be predictive of future development of clinical LMN signs within 6 months of identification and were identified within 4 years in 77% of patients who initially presented with a pure UMN phenotype [8]. Bulbar symptoms were more common in those with UMNdALS and ALS as compared to PLS patients across all visits and onset of disease in the bulbar region was also more common in UMNdALS (33%) and ALS (20%) as compared to PLS (11%) [15]. ‘
-8. Gordon, P.H., et al., The natural history of primary lateral sclerosis. Neurology, 2006. 66: p. 647-653.
-15. Gordon, P.H., et al., Clinical features that distinguish PLS, upper motor neuron-dominant ALS, and typical ALS. Neurology, 2009. 72: p. 1948-1952.
P3.1
‘Electromyography: this paragraph in particular should be better elaborated. Please better describe and discuss the EMG features that could be helpful in differentiating the three conditions, specifically referring to current EMG diagnostic criteria for ALS.’
-lines 211-236 expanded discussion to read ‘The initial study by Gordon et. al. that proposed the diagnostic category of UMNdALS defined clinically pure PLS as those who had a normal EMG of three limbs, bulbar, and paraspinal muscles while those with UMNdALS had evidence of lower motor neuron weakness, wasting, or fasciculations limited to one to two muscles or minor denervation on EMG, including sparse fibrillation/positive sharp waves, or minor motor unit potential remodeling in one to two muscles not meeting revised El Escorial criteria for any ALS category [8]. In the initial studies, EMG abnormalities occurred after a median of 3.17 years after development of UMN signs and preceded the LMN signs (atrophy, fasciculations, weakness) by six months. The clinical utility of identifying UMNdALS was to identify patients with UMN predominant disability that also had LMN signs as this predicted a prognosis, both in terms of survival and disability, that was intermediate as compared to PLS and typical ALS [8, 15]. While subsequent studies have reported no difference in survival between those with a diagnosis of PLS and a normal EMG and those with minimal denervation changes restricted to a single muscle, these studies did not include the presence of LMN signs in their definitions and limited the findings to a either ‘minimal’ EMG changes to a single muscle or fasciculations limited to no more than two muscles or fibrillation or motor unit changes in any muscle as qualifying as minor denervation on EMG [43, 44]. A prior study by Singer et.al. did note increased lower limb disability in those PLS patients with active denervation changes on their EMGs as compared to those patients with a normal EMG, although, the study was limited by a small sample of 25 total patients [12]. Taken together, if denervation changes are limited to a single or few muscles and are non-progressive or disappear on a repeat EMG at 4 years in an otherwise pure UMN syndrome, a diagnosis of definite PLS can be made. On the other hand, if a patient has evidence of denervation on their initial EMG but they develop limited/focal LMN symptoms (weakness, fasciculations, atrophy) over the course of 4 years but still do not meet revised El Escorial or Awaji criteria for ALS, a diagnosis of UMNdALS as opposed to probable PLS would be appropriate as they may have an intermediate prognosis as compared to typical PLS [4].’
-lines 237-238: corrected sentence to read ‘Various cut-offs for the development of lower motor neuron findings on EMG to distinguish PLS from UMNdALS or ALS have been used…’ by adding ‘or ALS’
P3.3
‘The presence of an hyperintensity of the subcortical white matter of the precentral gyrus or of the cortico-spinal tract is not specific of MND and can be discovered also in healthy patient (Ngai S, Tang YM, Du L, Stuckey S (2007). AJNR Am J Neuroradiol 28:250–254). This should be discussed.’
-lines 288-291: added ‘Despite its potential utility in predicting prognosis in upper motor neuron disorders, T2 CST+ may not be as useful from a diagnostic perspective as it has also been demonstrated in a normal population without evidence of motor neuron disease, specifically in those over the age of 50 [54].’ and included the recommended reference
‘ Even if the corpus callosum degeneration has been shown to be an earlier feature on neuroimaging, PLS patient showed a more severe damage to the motor corpus callosum fibers (Agosta F, et al (2014) Hum Brain Mapp 35:1710–1722. https://doi.org/10.1002/hbm.22286)’
-lines 301-308: added ‘Prior diffusion tensor imaging studies comparing PLS to ALS have supported the hypothesis of mid-body corpus callosum white matter changes in PLS reflecting secondary Wallerian degeneration as the result of neuronal death in the primary and premotor cortices. While there is similar distal CST involvement in PLS and ALS patients, those with PLS demonstrated more severe damage to the rostral portions of the CST as well as the motor callosal fibers as compared to ALS patients [56]. These findings support the recent PLS consensus classification system and argues for earlier intervention/inclusion into clinical trials [55, 57-59].’ and included recommended reference.
P 3.5.
‘In the paragraph about transcranial magnetic stimulation, I would also report that cortical inexcitability predominates in PLS, whereas in ALS the phenotype is more clearly one of cortical hyperexcitability (Geevasinga et al 2015).’
-Lines 328-335: Incorporated reference and modified text to state: ‘Threshold tracking transcranial magnetic stimulation techniques have demonstrated cortical dysfunction across motor neuron diseases which exists on a spectrum of cortical inexcitability as a predominant feature in PLS and cortical hyperexcitability as a predominant feature in ALS [27, 63, 64]. The stimulus intensity required to evoke a response has been shown to be higher in PLS as compared to ALS patients on average whereas cortical excitability is preserved in cases of HSP [64]. In addition to distinguishing between cases of PLS and ALS based on differences in cortical excitability, findings TMS threshold tracking techniques may also help differentiate PLS from HSP, which has persevered cortical excitability as compared to controls [64].’
P3.6
‘As reported in previous articles, NFL (and also pNFH) levels are higher in ALS with prevalent UMN involvement, suggesting that the rate of motor axon degeneration drives the difference in NF levels between ALS and PLS (Gaiani et al 2017; Falzone et al 2020).’
-Lines 341-355: Modified to include the above references and expand discussion ‘Both neurofilament light chains (NfL) and phosphorylated neurofilament heavy chains (pNfH) can be measured in serum and in CSF and have been shown to be correlated with disease severity and survival parameters in motor neuron disease [66, 67]. The mechanism underlying their elevation (UMN degeneration versus LMN degeneration) continues to be a matter of debate. One recent study speculated that elevated NfL in motor neuron disease was reflective of both the overall rate of motor neuron degeneration as well as the amount of degeneration in the anterior horn cells. This is supported by lower comparative values in pure UMN degeneration syndromes such as PLS or HSP as compared to classic ALS, although, diagnostic cut offs have yet to be clearly defined [68-71]. Alternatively, recent studies have found higher levels of CSF NfL and serum pNfH chains in typical ALS as well as those with UMNdALS and pseudobulbar palsy as compared to those syndromes that primarily affect the lower motor neurons such as progressive muscular atrophy or the flail arm/leg syndromes [66, 67]. These findings along with the known elevated levels (serum or CSF) of NfL or pNfH in ALS as compared to PLS or HSP may suggest that the rate of degeneration of the corticospinal tract drives the differences between these disorders [66, 67, 72, 73].’
P3.7
‘A recent autoptic series reported cortical TDP-43 pathology in 5 out of 5 PLS patients (Mackenzie et al 2020, TDP-43 pathology in primary lateral sclerosis). Moreover, the discovery of TDP-43 pathology also in LMN of PLS patients should be discussed and referenced (TDP-43 pathology in primary lateral sclerosis Ian R. A. Mackenzie & Hannah Briemberg).’
-Lines 368-382: added discussion of this study, first discussing limitations of prior clinicopathologic studies in PLS and finally discussing the TDP-43 pathological findings in PLS and similarities/differences with ALS ‘Several cases were primarily published based on novel clinical-pathologic associations due to their combination of upper motor neuron degeneration with atypical parkinsonian syndromes or common dementias such as FTD or dementia with lewy bodies, making them difficult to generalize to more ‘clinically pure’ cases of PLS [80-85]. A recent autopsy study of 7 cases that met 2020 consensus criteria for PLS demonstrated TDP-43 neuronal cytoplasmic inclusions (NCI) in the primary motor cortex of all 5 cases that it was available for analysis. Additionally, all cases demonstrated chronic degeneration of the corticospinal tract and no obvious reduction in the number of lower motor neurons in the hypoglossal nucleus or ventral gray matter but rare TDP-43 NCI in the LMNs. The number of LMN NCI in the PLS cases was estimated to be less than 1% as compared to an average of approximately 20% of cells demonstrating NCIs in ALS cases, including those designated as ‘early ALS’ [86]. While the frequency of TDP-43 NCI in the UMN of PLS may explain the association of FTD with both disorders, the significant difference in LMN TDP-43 NCI in the PLS cases as compared to ALS (both early and late) suggests that the disorders differ in their degree of LMN vulnerability [86].
-Table 2: line 396-added subscript to TDP-43 inclusion column for PLS that reads ‘TDP-43 NCI reported in primary motor cortex of PLS patients but rarely found in LMNs as compared to ALS’
P4.
‘The use of UMNdALS versus PLS and ALS remains difficult and perhaps the authors can critically analyse these groups ie whether UMNdALS is needed and why. Perhaps in the future directions this could be expanded?’
-Lines 403-406: added ‘Until such technology becomes more reliable, distinguishing PLS and ALS from UMNdALS remains important as this cohort of patients have been shown to have an intermediate progression in terms of disability and survival as compared to either PLS and ALS and may confound results of future clinical trials if they are included in PLS or ALS patient cohorts. ‘
Minor:
-Line 33- added ‘of’ (‘…and examination findings limited to the pyramidal tracts without evidence of in-volvement of additional parts of the central nervous system.)
-Line 49- corrected beginning of sentence to read ‘More’ as opposed to prior typo ‘ore’
-Line 241- added space (i.e. overtime to over time)
-Line 244- corrected EMGSs to EMGs
-Line 271- corrected ‘tarted’ to ‘targeted’ in ‘…gene targeted clinical trials…’
-Line 323- corrected PEG to PET
-Line 343-344- corrected ‘….can be measured both in serum and in blood…’ to ‘…can be measured both in serum and in CSF…’
We greatly appreciate the constructive feedback received on our initial draft and believe that with incorporation of these recommendations, this manuscript is now appropriate for publication by the Journal of Brain Sciences special issue Upper Motor Neurons in Health and Disease. This review provides an up to date summary of the findings in regards to clinical history/exam, EMG, genetics, pathology, and advanced neuroimaging for distinguishing the key upper motor neuron syndromes.
Thank you for your consideration!
Sincerely,
Timothy Fullam, M.D.
Maj, USAF, MC
Department of Neurology
University of Kansas Medical Center
Reviewer 2 Report
This is an interesting and timely focusing on the issue of Upper motor neuron disorders. Overall, the paper is well-written and organized and would be of interest for readers. However, in my opinion there are some points that need to be further clarified
P2.1. Line 82 Even if different onset sites do not differ for survival, a recent study compared a younger age at onset for paraparetic PLS sub-phenotype compared to bulbar and Mills phenotypes (Hassan et al 2021).
P2 Lines 82-87. Authors should provide a reference. Moreover, the less common upper limb onset should be commented, being widely more common in ALS than in PLS.
P2.2. Line 118. It would be useful to clear the point about “EMG denervation or LMN signs that do not yet meet criteria for ALS”, referring to revised El Escorial and/or Awaji criteria.
P2 Lines 121-126. Please provide a reference.
P 2.1: extra motor features such as parkinsonism should also be discussed (e.g. Parkinsonism and PLS: norlinah iM, Bhatia KP, Østergaard K, et al. Primary lateral sclerosis mimicking atypical parkinsonism. Mov Disord 2007;22:2057–62; Mabuchi n, Watanabe h, atsuta n, et al. Primary lateral sclerosis presenting parkinsonian symptoms without nigrostriatal involvement. J Neurol Neurosurg Psychiatry 2004;75:1768–71)
P3 Lines 249-258. The presence of an hyperintnsity of the subcortical white matter of the precentral gyrus or of the cortico-spinal tract is not specific of MND and can be discovered also in healthy patient (Ngai S, Tang YM, Du L, Stuckey S (2007). AJNR Am J Neuroradiol 28:250–254). This should be discussed.
P3 Lines 267-269. Even if the corpus callosum degeneration has been shown to be an earlier feature on neuroimaging, PLS patient showed a more severe damage to the motor corpus callosum fibers (Agosta F, et al (2014) Hum Brain Mapp 35:1710–1722. https://doi.org/10.1002/hbm.22286)
3.1 Electromyography: this paragraph in particular should be better elaborated. Please better describe and discuss the EMG features that could be helpful in differentiating the three conditions, specifically referring to current EMG diagnostic criteria for ALS.
P 3.5. In the paragraph about transcranial magnetic stimulation, I would also report that cortical inexcitability predominates in PLS, whereas in ALS the phenotype is more clearly one of cortical hyperexcitability (Geevasinga et al 2015).
P3.6 Line 305. As reported in previous articles, NFL (and also pNFH) levels are higher in ALS with prevalent UMN involvement, suggesting that the rate of motor axon degeneration drives the difference in NF levels between ALS and PLS (Gaiani et al 2017; Falzone et al 2020).
P3.7: a recent autoptic series reported cortical TDP-43 pathology in 5 out of 5 PLS patients (Mackenzie et al 2020, TDP-43 pathology in primary lateral sclerosis). Moreover, the discovery of TDP-43 pathology also in LMN of PLS patients should be discussed and referenced (TDP-43 pathology in primary lateral sclerosis Ian R. A. Mackenzie & Hannah Briemberg).
Minors:
Line 209: “EMGSs”
Line 239: “gene tarted clinical trials”
Line 302: “both in serum and in blood” I guess you mean both in serum and in CSF
Author Response

(The authors gave the same response as above.)

Round 2
Reviewer 1 Report
no additional suggestions